# Effect of Socioeconomic Status on Altruistic Behavior in Chinese Middle School Students: Mediating Role of Empathy

**DOI:** 10.3390/ijerph20043326

**Published:** 2023-02-14

**Authors:** Xiaomin Liu, Yuqing Zhang, Zihao Chen, Guangcan Xiang, Hualing Miao, Cheng Guo

**Affiliations:** 1Research Center of Mental Health Education, Faculty of Psychology, Southwest University, Chongqing 400715, China; 2School of Psychology and Cognitive Science, East China Normal University, Shanghai 200062, China; 3Tian Jiabing College of Education, China Three Gorges University, Yichang 443002, China

**Keywords:** socioeconomic status, altruistic behavior, empathy, cognitive empathy, affective empathy

## Abstract

Previous studies have shown that socioeconomic status is correlated to altruistic behavior. The role of empathy as one of the motivations for altruistic behavior is gradually gaining attention among researchers. This study explores the role of empathy in the mechanisms of socioeconomic status and altruistic behavior in Chinese adolescents. A total of 253 middle school students from Northern China participated in this study, which included the dictator game and Interpersonal Relation Index. Results showed that (1) low-SES students behaved more generously than high-SES students; (2) the students were more generous to the low-SES recipients, as shown when offering them more money in the dictator game; (3) affective rather than cognitive empathy mediates the relationship between socioeconomic status and altruistic behavior. The findings provide evidence for the validation of the empathy–altruism hypothesis in a group of Chinese adolescents. Meanwhile, it reveals the path to improving altruistic behavior through the promotion of empathy, especially for individuals of high socioeconomic status.

## 1. Introduction

Altruistic behavior is defined as paying a cost to promote the well-being and interests of others, without seeking any benefit in return [1,2,3]. Prior studies provide direct support for the prediction that altruistic behavior can promote physical health. For example, individuals who performed altruistic behaviors (helping and donating) reported a warmer perception than those who did not and were better able to withstand a cold environment [4]. In addition, the behavioral evidence suggested that altruistic behavior was associated with more self-reported positive effects [5]. Despite the sacrifice of self-interest, altruistic behavior often promotes the altruist’s sense of self-efficacy and brings rewards such as reputation and reciprocity [6,7,8]. In addition to the psychological evidence, the cooperative spirit of economics allows individuals to increase their altruistic behavior to achieve social prosperity [9]. Therefore, altruistic behavior is an indicator of individual physical and mental health, and social well-being and is significant for social prosperity [10,11]. Adolescence is a period of rapid emotional and cognitive development. Adolescents gain more opportunities to engage in altruistic behaviors due to changes in the social environment and interpersonal relationships. It is essential to explore the potential influencing factors that account for altruistic sharing behavior in adolescents.

### 1.1. Socioeconomic Status and Altruistic Behavior

Socioeconomic status (SES), also known as social class, refers to the differences in objective social resources (income, education, and occupation) as well as subjective differences in perceived relative position between groups in different positions in the social hierarchy [12,13]. Evidence from previous empirical research indicates that socioeconomic status is negatively related to altruistic behavior [14,15,16]. The prior research found that individuals of lower socioeconomic status are more prosocial than those of higher socioeconomic status, such as donating a higher proportion of their income to charity or sharing more in the dictator game [17,18]. The dictator game is one of the typical paradigms for laboratory studies of altruistic behavior, in which the amount of money given by a participant while playing the role of a dictator is seen as his or her level of altruistic behavior [19,20]. This tendency exists not only in the laboratory but also in the real situation, as lower-socioeconomic-status individuals show more intention and behavior to share their wealth [18,21,22]. Kraus proposed the social cognitive theory of social class wherein the high empathy and communal self-concept of individuals with low socioeconomic status make them more altruistic [12]. Although the research on SES and altruistic behavior is abundant, there are few studies focused on Chinese adolescents. Teenagers are still in a stage of economic confusion, so whether the conclusions of previous studies can be applied remains to be further tested. In addition, a study by Amir and Rand found that disadvantaged early-life experience such as low socioeconomic status predicts greater risk-taking, more present time orientation, and more prosociality in adulthood [21]. Therefore, investigating the mechanisms underlying the relationship between teenagers’ socioeconomic status and altruistic behavior is conducive to guiding pre-adult growth.

### 1.2. Empathy and Altruistic Behavior

Empathy is the ability to put oneself in the situation of feeling and understanding the emotions and thoughts of others [23]. It refers to the state in which an individual recognizes and experiences the emotions of others in a given situation, and is a reflection of the individual’s ability to recognize and experience others’ emotions, as well as the process of sharing, understanding, and responding to others’ emotions [24]. Most researchers consider that empathy consists of two main components: cognitive and affective empathy [25,26,27]. Cognitive empathy focuses on the reasoning and judgment of emotional states, while affective empathy is mainly about the feeling and experience of others’ emotional states [25,28]. Therefore, affective empathy can be seen as a deeper extension of cognitive empathy, which is an empathic emotional response that results from reasoning about an emotional state [28].

Empathy is seen as one of the motivations for altruistic sharing behavior. Batson proposed the empathy–altruism hypothesis, which suggests that when others experience difficulties, observers generate an emotion directed toward the recipient, including empathy, sympathy, and compassion [29]. The greater the intensity of this emotion, the stronger the altruistic motivation of the observers to relieve the difficulties of others, and the more likely he or she is to engage in helping behavior. Empathy has been found to help individuals to become more attentive to the feelings and needs of others, inspiring prosocial behaviors such as altruism, helping, and donating [29,30,31,32]. Additionally, empathy predicts altruistic behavior in the dictator game; individuals with high levels of empathy behave more generously. Klimecki and colleagues found that dictators who watched videos of suffering others had higher levels of self-reported empathy and shared more money with the recipients than control group dictators [33]. In addition, researchers found that cognitive empathy is associated with people’s altruistic behavior [34]. Eisenberg and Miller proposed that affective empathy is more relevant to altruistic behavior than cognitive empathy [35]. Gummerum and Hanoch compared the effects of cognitive empathy and affective empathy on sharing behavior in adults and found that only affective empathy is positively correlated with the dictators’ sharing amount [36].

### 1.3. Socioeconomic Status and Empathy

Based on the social cognitive theory of social class, different levels of socioeconomic status elicit social cognition differently, such as empathy [12]. For example, research demonstrates that both objective and subjective socioeconomic status are positively related to empathic accuracy [16]. Lower-socioeconomic-status individuals were more accurate in their empathizing with others’ emotions than higher-socioeconomic-status individuals. The result is interpreted in terms of lower-socioeconomic-status individuals being more inclined to explain social events based on features of the external environment and more capable of identifying others’ emotional states. Moreover, research revealed that the objective socioeconomic status, composed of household income and educational attainment, could negatively predict empathy performance [16]. Evidence from Neuroimage suggested that P2 responses, an event potential (ERP) indicator associated with empathy, are negatively correlated with socioeconomic status, as reflected by the reduced neural empathic response in high-social-class individuals [37].

### 1.4. Mediating Role of Empathy 

The social cognitive theory of social class reveals that individuals with low socioeconomic status exhibit higher levels of empathy, and the empathy–altruism hypothesis suggests that high empathy promotes altruistic behavior. Therefore, based on previous research and theory, we infer that empathy may play a mediating role between socioeconomic status and altruistic sharing behavior. A few studies have given indirect evidence for this supposition. For instance, researchers found that high-socioeconomic-status individuals showed less prosocial behavior in contrast with low-socioeconomic-status individuals, while watching a compassion-inducing video could increase their prosocial behavior [18]. In interpreting the results that high-socioeconomic-status individuals engage in less wealth-sharing behavior, Lim and Desteno suggested that this may stem from their low level of empathy, resulting in insensitivity and inattention to the needs and interests of others [38]. Barraze and Zak found that the experience of empathy raises oxytocin levels and promotes the generosity of sharing with strangers [39].

In addition, the essence of altruistic behavior is a social activity that occurs between the helper and the receiver. Therefore, researchers have found that the identity of the recipient is an influential factor in altruistic behavior [15,40]. Individuals are more prosocial toward poor recipients than wealthy ones [40,41]. Furthermore, based on the theory of group identity, people are often divided into in-groups and out-groups [42]. Group identity drives in-group favoritism and intergroup bias. In-group favoritism motivates individuals to engage in more altruistic and cooperative behaviors toward in-group members [43,44,45]. Accordingly, we infer that more altruistic behavior would result if the socioeconomic status of the helper and the recipient were aligned (both low or high). The presence of in-group favoritism and intergroup bias may lead to an interaction between the socioeconomic status of the helper and the recipient in terms of altruistic behavior. Given that high-socioeconomic-status recipients cannot elicit the helpers’ empathy, we consider that the model “empathy mediates the relationship between the socioeconomic status of the helper’s and altruistic behavior” is valid only when faced with low-socioeconomic-status recipients.

### 1.5. The Current Study

The current study aims to provide more insight into the mechanism underlying the association between socioeconomic status and altruistic behavior and reveal the specific role of empathy in this relationship. Based on the previous literature review, we propose the following three hypotheses.

**Hypothesis** **1** **(H1).***The helper’s socioeconomic status is negatively related to altruistic behavior, and empathy is positively related to altruistic sharing behavior*.

**Hypothesis** **2** **(H2).***There is an interaction between the socioeconomic status of the helper and the recipient in altruistic behavior. The degree of negative correlation between the helper’s socioeconomic status and altruistic behavior is more sloped when faced with a recipient of low socioeconomic status compared to a recipient of high socioeconomic status*.

**Hypothesis** **3** **(H3).***Empathy mediates the relationship between socioeconomic status and altruistic behavior when the recipient is of low socioeconomic status (see Figure 1)*.

## 2. Methods

### 2.1. Participants

Participants were filtered by a questionnaire on their family monthly income from a middle school in Northern China. Students’ annual family income (with CNY 1000 as the smallest unit) was measured previously in another study on this school. Based on the survey data, we knew that the top 27% and bottom 27% of the overall annual income of the sample was 30,000 yuan and 150,000 yuan, respectively (in Chinese yuan, CNY 1 = USD 0.14 when the survey was conducted). Therefore, we defined the income criteria for grouping low-objective-socioeconomic-status and high-objective-socioeconomic-status families. Students whose family income was less than 2500 yuan per month were classified into the low-objective-socioeconomic-status group (low-SES group, lses for short), while the ones whose family income was more than 15,000 yuan per month were classified into the high-objective-socioeconomic-status group (high-SES group, hses for short).

We obtained school approval, as well as participants’ own and their parents’ informed consent, before this study. Participants were compensated with small gifts (e.g., candies or pencils) for completing this survey. We used G * power (Version 3.1) to determine the sample size of this study [46]. Social science research has always suggested a moderate effect size (*r* = 0.40) for the association between socioeconomic status and altruistic behavior. We needed at least 142 participants, which could provide 95% statistical power to detect a moderate effect. In this study, a total of 796 student participants from 15 classes in 3 grades, with 5 classes in each grade, were sampled using a convenience sampling method. There were 125 participants in the low-SES group and 128 participants in the high-SES group, for a total of 253 participants included in this study, aged from 11 to 16 years old (*M*_age_ = 12.58 years, *SD* = 0.98), and 44.66% were male. 

### 2.2. Procedure and Materials

After giving their informed consent, participants first reported their age and gender. Then, they completed the dictator game. After finishing the dictator game, participants filled in the Interpersonal Reactivity Index Scale.

#### 2.2.1. Altruistic Behavior

Altruistic behavior was assessed through the amount of money that the dictators offered to the receivers in the dictator game [19,20]. Two players participate in this game, who are assigned as the dictator and the recipient. They are given a quantity of money or goods, and the dictator decides upon the amount of money or goods to be allocated to the recipient, while the recipient can only accept the allocation result unconditionally [47]. The amount allocated by the dictator is regarded as the level of the dictator’s altruistic sharing behavior. 

In this study, researchers offered 10 yuan to the dictator and recipient. Participants played the role of a dictator and distributed money (any integer from 0 to 10 yuan) to another middle school student. Each participant finished three trials for three levels of the recipient’s socioeconomic status (undefined condition/poor-recipient condition/wealthy-recipient condition). The procedure is presented by E-prime 2.0. The first trial was the undefined condition, in which the recipient was described as an unknown middle school student without any additional description. The recipient in the following two trials was described to be a poor or a wealthy unknown student with social cues—for example, he or she wears clothes that are washed white or shoes that are worn with holes, or he/she is wearing designer clothes and shoes. The second and third trials were presented randomly. To avoid the influence of reciprocity and social approval effects on the experimental results, we informed participants that their arrangements were anonymous and that the recipients were different in all three trials. After the last two trials, as a manipulation check, participants were required to rate the economic status of the recipient’s family using a 10-point scale (1 = the worst, 10 = the best).

#### 2.2.2. Empathy, Cognitive Empathy, and Affective Empathy 

Participants completed a measure of perspective taking and empathic concern via the Chinese version of the Interpersonal Reactivity Index (IRI), the most commonly used instrument to measure empathy. The scale was originally developed by Davis, consisting 28 items in 4 dimensions: perspective taking, empathic concern, fantasy, and personal distress [28]. Among these subscales, the perspective taking subscale measures the tendency to spontaneously adopt the psychological point of view of others and is commonly used to measure cognitive empathy [28]. The empathic concern subscale measures the individual’s feelings of compassion and concern for others in unfortunate circumstances and is used to indicate affective empathy [25,28]. In the simplified Chinese version of IRI, there are 5 items on perspective taking and 6 items on empathic concern, such as “Before criticizing others, I consider how I would feel if I were in their position” and “For those who are not as fortunate as I am, I often feel thoughtful and concerned”. Participants reported the extent to which they agreed with the discrimination of the items on a five-point Likert scale (1 = do not agree at all, 5 = absolutely agree). The scores of the two subscales represent cognitive empathy and affective empathy, respectively, and the sum of the two scores represents the overall empathy of the participants, with higher scores indicating higher empathic ability. In the current study, the scale showed good reliability (Cronbach’s *α* of all the items is 0.77, with Cronbach’s α of 0.80 for the subscale of perspective taking and 0.72 for the subscale of empathic concern). 

### 2.3. Data Analysis

SPSS 22.0 was used to analyze the data. Firstly, manipulation checks were conducted, including whether participants in the low-SES group and high-SES group differed in their demographic variables and whether there was a significant difference in participants’ perceptions of the socioeconomic status of the wealthy recipient and poor recipient. Secondly, descriptive analysis and Pearson’s correlations were performed regarding cognitive empathy, affective empathy, empathy (the arithmetic means of cognitive empathy and affective empathy), and altruistic sharing behavior. Then, the moderating effect of the recipient’s socioeconomic status on sharing was tested. Finally, the PROCESS Macro was used to test the mediating effect of empathy, cognitive empathy, and affective empathy in detail, between the dictator’s objective socioeconomic status and altruistic sharing behavior.

## 3. Results 

### 3.1. Manipulation Check 

An independent-sample *t* test was performed on the demographic composition of the two groups of subjects before data analysis. The results revealed no significant differences between the two groups of subjects in terms of age and gender composition (*p* = 0.23; *p* = 0.65) (see Table 1).

A paired-sample *t* test of the perception of the recipient’s socioeconomic status, which participants rated based on the social cues of the description, was conducted. The score for the poor recipient was 2.87 and the score for the wealthy recipient was 6.95 (*t* = 18.24, df = 252, *p* < 0.001). This indicates that the descriptions containing social cues caused subjects to feel that the poor recipients were of a lower socioeconomic status than the wealthy recipients. Thus, the socioeconomic status of recipients’ conditions was manipulated successfully. 

### 3.2. Descriptive Analyses and Correlations

Descriptive statistics and bivariate correlations are shown in Table 2. The mean score of overall empathy was 3.80 (*SD* = 0.59), and the mean scores for the two dimensions of empathy were as follows: the score for cognitive empathy (perspective taking) was 3.96 (*SD* = 0.72), and that for affective empathy(empathic concern) was 3.68 (*SD* = 0.66). As hypothesized, empathy was positively correlated with altruistic sharing behaviors under the three conditions (*r* = 0.28, *p* < 0.001; *r* = 0.30, *p* < 0.001; *r* = 0.21, *p* < 0.01). In addition, both cognitive empathy and affective empathy were positively correlated with altruistic sharing behaviors significantly (see Table 2).

### 3.3. The Moderating Role of Recipient’s Socioeconomic Status

The objective socioeconomic status (SES, low/high) of the dictator was the between-subjects independent variable, the socioeconomic status (poor/wealthy/undefined) of the recipient was the within-subjects independent variable, and the amount of money allocated by the dictator to the recipient in the dictator game was the dependent variable. Subsequently, a 2 (dictator’s objective socioeconomic status: low SES/high SES) × 3 (recipient’s socioeconomic status: poor/wealthy/undefined) repeated-measures ANOVA test was conducted on the amount of altruistic sharing allocation. The results revealed a significant main effect of the dictator’s objective socioeconomic status, *F* (1,251) = 5.00, *p* = 0.026, *η*^2^ = 0.02. The allocation was higher in the low-SES group (*M* = 5.26, *SD* = 1.56) than in the high-SES group (*M* = 4.95, *SD* = 1.54). The main effect of the recipient’s socioeconomic status also reached a significant level, *F* (2,250) = 325.24, *p* < 0.001, *η*^2^ = 0.72. Subjects’ allocation was lower when faced with a wealthy recipient (*M* = 3.42, *SD* = 1.83) than in the undefined-recipient condition (*M* = 4.67, *SD* = 0.99) and poor-recipient condition (*M* = 7.24, *SD* = 1.89). In addition, the interaction between the dictator’s objective socioeconomic status and the recipient’s socioeconomic status was significant, *F* (2,250) = 3.98, *p* = 0.02, *η*^2^ = 0.02. Furthermore, a simple effect analysis revealed that the allocations to undefined and poor recipients showed significant differences in the low-SES group and high-SES group (undefined-recipient condition: *M _lses_* (4.78) > *M _hses_* (4.55), *p* = 0.057; poor-recipient condition: *M _lses_* (7.59) > *M _hses_* (6.89), *p* = 0.003). However, in the wealthy-recipient condition, there was no significant difference in the allocation of the two groups, *M _lses_* = 3.41, *M _hses_* = 3.42, *p* = 0.952 (see Figure 2a).

Additionally, the allocation amount in the undefined condition was set as the allocation expectation. Setting the expectation as a baseline, the amount of change was formed by subtracting the allocation expectation from the allocation amount when subjects faced poor and wealthy recipients. Δ_1_ is a positive increase towards the poor-recipient condition and Δ_2_ is a negative decrease in the wealthy-recipient condition (see Figure 2b). A repeated-measures ANOVA of 2 (dictator’s objective socioeconomic status: low SES/high SES) × 2 (recipient’s socioeconomic status: poor/wealthy) was conducted for Δ_1_ and Δ_2_. Results showed a significant main effect of the recipient’s socioeconomic status, *F* (1,251) = 572.91, *p* < 0.001, *η*^2^ = 0.70, with the amount given to the poor recipient (Δ_1_) (*M* = 2.58, *SD* = 1.73) being significantly larger than the amount given to the wealthy recipient (Δ_2_) (*M* = −1.25, *SD* = 1.75). However, the main effect of the dictator’s objective socioeconomic status was not significant, *F* (1,251) = 0.50, *p* = 0.48, *M _lses_* = 0.72, *M _hses_* = 0.61. The interaction between the dictator’s objective socioeconomic status and the recipient’s socioeconomic status was significant, *F* (1,251) = 5.00, *p* = 0.026, *η*^2^ = 0.02. A simple effect analysis was conducted and discovered that the amount allocated to the poor recipient (Δ_1_) was significantly different between the low-SES group and high-SES group, *M _lses_* (2.81) > *M _hses_* (2.34), *p* = 0.035; the reduction for wealthy dictators (Δ_2_) was not significantly different between the two groups, *M _lses_* = −1.38, *M _hses_* = −1.13, *p* = 0.255.

### 3.4. The Mediating Effects of Cognitive Empathy and Affective Empathy

We conducted a mediation analysis to test whether empathy mediated the relation between the objective SES of the dictator (0 = *low SES*, 1 = *high SES*) and altruistic sharing behavior in the poor- and wealthy-recipient conditions using a bootstrapping procedure with 5000 resamples [48]. We tested our hypotheses using the PROCESS Macro (Model 4) in SPSS 22.0. All continuous variables were standardized before the analysis [49]. 

In the poor-recipient condition, the significant effect of the dictator’s SES on altruistic sharing behavior (*β* = −0.37, *p* = 0.003) became nonsignificant (*β* = −0.22, *p* = 0.08) after including empathy in the model. Moreover, the indirect effect of empathy was significant, indirect effect = −0.41, 95%CI [−0.52, −0.11], accounting for 41.43% of the total effect (see Table 3, Figure 3a).

Given that affective empathy, which is measured by empathic concern, is more relevant to the situation of economic games, we speculated that the two sub-components of empathy played different roles in the mediation model. Further, we tested the specific hypotheses about the parallel mediation effects of cognitive empathy and affective empathy using Model 4 in the PROCESS Macro (Model 4) in SPSS 22.0 [49]. According to the results, the mediating effect of cognitive empathy was not significant (95% CI [−0.21, 0.11]); the mediating effect of affective empathy was significant (95% CI [−0.47, −0.09]), accounting for 35.71% of the total effect (see Table 4, Figure 3b). Thus, it can be seen that the dictator’s SES negatively predicts altruistic sharing behavior when faced with a poor recipient, partly through the mediating effect of affective empathy rather than cognitive empathy.

Although we did not formulate a specific hypothesis about the role of empathy in mediating the relationship between the dictator’s SES and altruistic sharing behavior in the wealthy-recipient condition, we tested the possibility. The results demonstrated that the dictator’s SES could not predict altruistic sharing behavior (*β* = 0.14, *p* = 0.28, 95% CI [−0.21, 0.72]), while empathy positively predicted altruistic sharing behavior (*β* = 0.23, *p* < 0.001, 95% CI [0.32, 1.11]). The mediation model did not work overall (*F* (1251) = 0.004, *p* = 0.95). Thus, empathy did not mediate the relation between the dictator’s SES and altruistic sharing behavior in the wealthy-recipient condition.

## 4. Discussion

The results show that the dictator’s objective socioeconomic status was negatively correlated to sharing behavior, which supports Hypothesis 1. This may due to the self-oriented psychological tendencies of individuals with high socioeconomic status [12]. It refers to the mode of thinking and behavior in which individuals tend to be guided by their interests and preferences [13,38,50]. Individuals with high socioeconomic status have more wealth and resources and can achieve their personal goals independently, with less dependence on others; thus, they are more likely to form self-orientation psychological tendencies and perform less prosocial behavior [12,13]. On the contrary, based on the social cognitive theory of social class, the communal self-concept of individuals with low socioeconomic status makes them more altruistic [16]. 

Meanwhile, the moderating effect of recipient socioeconomic status is significant, which supports Hypothesis 2, with dictators sharing more money with poor recipients than with rich ones. Although the definition of altruistic behavior mentions that the helpers aim to increase the benefits of the recipients, researchers have found that individuals evaluate the needs of the recipients before engaging in altruistic behavior. When the recipients’ needs are sufficiently high, the individual will suppress their self-interested motives and thus engage in altruistic behavior [51,52]. Therefore, the dictators perceive the higher level of demand for money of the poor recipients and engage in more sharing behavior. 

The results also revealed an interaction between the socioeconomic status of dictators and recipients in sharing behavior. The simple effects analysis revealed that low-SES dictators showed more sharing behavior when faced with poor recipients; meanwhile, in the wealthy-recipient condition, there was no significant difference in the sharing behavior between low-SES and high-SES dictators. Combined with the analysis of the Δ_1_, the former part of the results can be explained by the high empathy and in-group favoritism of low-SES dictators, both of which promote sharing behavior, and the low empathy and intergroup bias of high-SES dictators, both of which weaken sharing behavior. Low-SES dictators showed in-group favoritism to poor recipients who were also of low SES. Moreover, similar experiences of difficulty make low-SES dictators more empathetic to poor recipients’ plights and more concerned about their interests, and they are therefore willing to share more money. This explanation was also verified in the following mediation examination. The promotion of altruistic behavior by empathic experience is consistent with the research by Piff and provides evidence for the empathy–altruism hypothesis in the Chinese adolescent population [18]. When faced with wealthy recipients, the dictators cannot demonstrate an empathic attitude, regardless of whether they are of low SES or high SES. Combined with the analysis of the Δ_2_, it is possible to see that both the intergroup bias of low-SES dictators and the in-group favoritism of high-SES dictators may not play a role in this. The results of the mediation examinations suggest that the mediating role of empathy exists only in the poor-recipient condition. 

Furthermore, we attempted to distinguish the mediating role played by the two components of empathy, cognitive and affective empathy, in the mechanism. Results revealed that it was affective empathy (empathic concern) rather than cognitive empathy (perspective taking) that mediated the effect. This finding supports Hypothesis 3 and is consistent with previous research showing that affective empathy is more associated with altruistic behavior than cognitive empathy [35,36]. This may be related to the paradigm of altruistic behavior research used in this study. In the dictator game, the dictator has complete power of allocation, while the receiver can only passively accept whatever the allocation is. At the same time, the dictator game has a strong contextual dimension. Therefore, the poverty of the recipients is more likely to initiate empathic concern rather than the perspective taking of the dictators. Under complete allocative power, emotional experiences can influence dictators to engage in altruistic behavior more directly than cognitive processes.

## 5. Conclusions

This study explored the relationship between objective socioeconomic status and altruistic sharing behavior in Chinese middle school students. It revealed that the participant’s objective socioeconomic status could negatively predict sharing behavior; students of low objective socioeconomic status behave more generously than those of high objective socioeconomic status. In the poor-recipient condition, empathic concern mediated the relationship between the participant’s objective socioeconomic status and altruistic sharing behavior.

Previous studies have focused on the influence of the helper’s socioeconomic status on altruistic behavior. The current study provides new research perspectives on the recipient’s socioeconomic status and the mediating role of empathy. The results suggest that situation-related affective empathy can facilitate altruistic behavior directed at helping others in need or distress. The findings provide evidence for the validation of the empathy–altruism hypothesis in a group of Chinese adolescents. The mediating role of empathy may provide some guidance and suggestions on how to improve altruistic behavior among high-socioeconomic-status individuals. For example, broadcasting public service announcements on disadvantaged people can inspire empathy among high-socioeconomic-status individuals and promote their sharing behavior. Given that studies showed that adverse experiences such as low socioeconomic status during childhood and adolescence can lead individuals to exhibit risky behavior and present a time orientation in adulthood, the finding of the mediating role of empathy in this study suggests that parents or teachers can intervene to train middle school students in empathy, so as to reduce such negative effects in the future. Further studies could better explore the mechanisms at play by manipulating or measuring in-group favoritism and out-group bias in such a way that separates group relations from socioeconomic status relations. 

## Figures and Tables

**Figure 1 ijerph-20-03326-f001:**
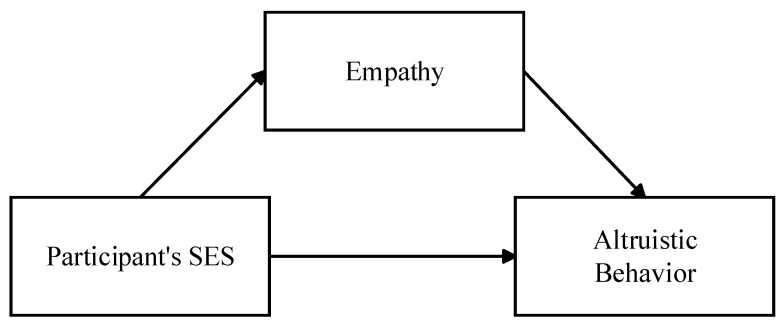
Hypothesized model.

**Figure 2 ijerph-20-03326-f002:**
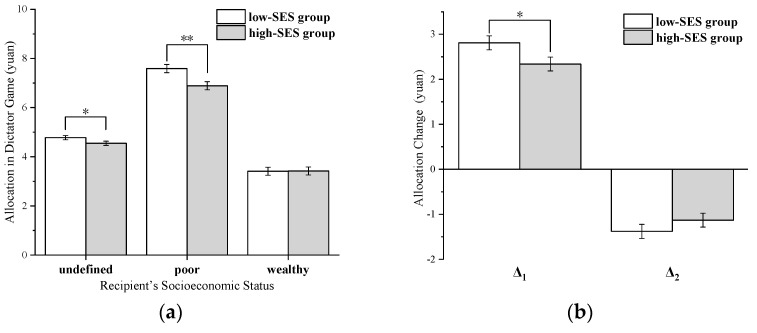
(**a**), Allocation of the dictators in the dictator game in three recipient conditions. (**b**) Allocation changes in two groups, Δ_1_ = allocation_poor_ − allocation_undefined_, Δ_2_ = allocation_wealthy_ − allocation_undefined_. * *p* < 0.05; ** *p* < 0.01.

**Figure 3 ijerph-20-03326-f003:**
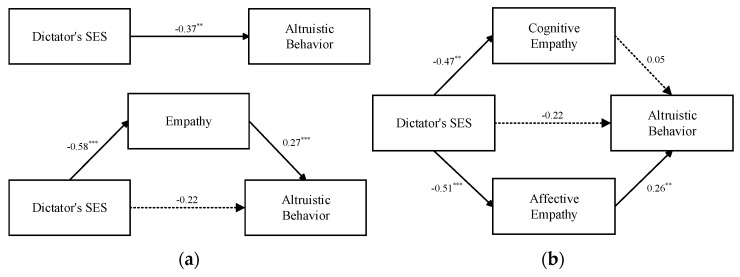
(**a**) Empathy as the mediator of the association between dictator’s SES and altruistic behavior in poor-recipient condition. (**b**) Cognitive empathy and affective empathy as mediators of the association between dictator’s SES and altruistic behavior in poor-recipient condition. Note: Dictator’s SES: 0 = low SES, 1 = high SES. ** *p* < 0.01. *** *p* < 0.001.

**Table 1 ijerph-20-03326-t001:** Age and gender composition of subjects in two groups.

	*N*	Age		Gender	
		M	SD	Male	Female
low-SES group	125	12.66	0.99	54	71
high-SES group	128	12.51	0.97	59	69

**Table 2 ijerph-20-03326-t002:** Means, SDs, and correlations among empathy, cognitive empathy, affective empathy, and sharing (*N* = 253).

	*M* ± *SD*	1	2	3	4	5	6
1. Empathy	3.80 ± 0.59	1					
2. Cognitive empathy	3.96 ± 0.72	0.83 ***	1				
3. Affective empathy	3.68 ± 0.66	0.87 ***	0.45 ***	1			
4. Sharing in urc	4.66 ± 0.99	0.28 ***	0.19 **	0.27 ***	1		
5. Sharing in prc	7.24 ± 1.89	0.30 ***	0.19 **	0.31 ***	0.39 ***	1	
6. Sharing in wrc	3.42 ± 1.83	0.21 **	0.18 **	0.18 **	0.35 ***	0.05	1

Note: 1. Empathy = arithmetic mean of cognitive empathy and affective empathy; 2. Cognitive Empathy = perspective taking; 3. Affective Empathy = empathic concern; 4. Sharing in urc = sharing in undefined-recipient condition; 5. Sharing in prc = sharing in poor-recipient condition; 6. Sharing in wrc = sharing in wealthy-recipient condition. ** *p* < 0.01. *** *p* < 0.001.

**Table 3 ijerph-20-03326-t003:** Regression coefficients, standard errors, and effects for the hypothetical mediation model (empathy as mediator).

	Effect Size	SE	Boot CI Lower	Boot CI Upper	Effect Proportion
Total effect	−0.70	0.23	−1.16	−0.26	
Direct effect	−0.29	0.11	−0.52	−0.11	41.43%
Indirect effect of empathy	−0.41	0.22	−0.85	0.03	58.57%

**Table 4 ijerph-20-03326-t004:** Regression coefficients, standard errors, and effects for the hypothetical mediation model (perspective taking and empathic concern as mediators).

	Effect Size	SE	Boot CI Lower	Boot CI Upper	Effect Proportion
Total effect	−0.70	0.23	−1.16	−0.26	
Direct effect	−0.41	0.22	−0.87	0.05	58.57%
Indirect effect of cognitive empathy	−0.04	0.08	−0.21	0.11	5.71%
Indirect effect of affective empathy	−0.25	0.10	−0.47	−0.09	35.71%

## Data Availability

Due to the privacy of the participants, the data will not be disclosed temporarily. If necessary, please contact the corresponding author.

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
