# Peer review of "Effect of Socioeconomic Status on Altruistic Behavior in Chinese Middle School Students: Mediating Role of Empathy"

_ijerph, 2023, doi:10.3390/ijerph20043326_

Round 1
Reviewer 1 Report
It was a great pleasure to read the manuscript entitled "Effect of Socioeconomic Status on Altruistic Behavior in Chinese Middle School Students: Mediating Role of Empathy," which I find well-written. Although the author/s strived to keep the quality of the study, I have a few comments I would like to forward to restore the soundness of the study.
1. Page 3. Like 146. "Hypothesis 2 (H2). There is an interaction between the socioeconomic status of the helper and the recipient in altruistic behavior" as long as the author/s stated this statement as speculation or hypothesized, it is necessary to rewrite the degree or direction of the relationship between the socioeconomic status of the helper and the recipient in altruistic behavior.
2. In the method section, I strongly suggest the author/s justify the reason behind or the rationale for conducting the study in Middle school.
3. Brief us about the sampling technique and methods.
4. After showing several significant findings and conclusions, the authors should forward policy, theoretical, and practical implications.
Reviewer 2 Report
The manuscript “Effect of Socioeconomic Status on Altruistic Behavior in Chi- 2 nese Middle School Students: Mediating Role of Empathy“ is very interesting research, but there are some minor comments.
Sentence in Introduction part seems irrelevant for this topic „Prior studies provide direct 27
support for the prediction that altruistic behavior can promote physical health. For exam- 28
ple, individuals who are in physically threatening situations exhibited relieving painful 29
feelings, when acting altruistically [4].
Please, write some reference for sentence: Cognitive empathy focuses on the reasoning and judgment of emotional states, while 67
affective empathy is mainly about the feeling and experience of others' emotional states. 68 Therefore, affective empathy can be seen as a deeper extension of cognitive empathy, 69
which is an empathic emotional response that results from reasoning about an emotional 70
state“.
It is not usual to put instrument description in Introduction part. For example sentence „Davis developed the most commonly used Interpersonal Relation Index (IRI), a 28- 71 item scale that measures cognitive empathy and affective empathy on four dimensions: 72 Perspective Taking (PT), Fantasy (FS), Personal Distress (PD), and Empathic Concern (EC) 73 [28]. Among these subscales, perspective taking sub-scale measures the tendency to spon- 74 taneously adopt the psychological point of view of others [28]“. So, please remove this sentences from Intordiction!
In Figure 1. Hypothesized model, please, change „Dictator's SES“with „participant SES“.
Authors write: „Participants reported how they agree with the discrimination of the 208
items on a five-point Likert scale (1=do not agree at all, 5=absolutely agree), with higher scores 209 indicating the higher empathic ability of the participants. In the current study, the scale 210
showed good reliability (Cronbach’s α = 0.77).“ What about reliability for subscales? Please, state more clearly what the measures of empathy are.
From this sentence: „Sec- 216
ondly, descriptive analysis and Pearson’s correlations were performed about empathy, 217
perspective taking, empathic concern, and altruistic sharing behavior. „ it is unclear if there are three measures for empathy (empathy, perspective taking, empathic concern), or perspective taking and empathic concern are indicators of empathy. Please clarify!
In Table 2.it is unclear what number stands for which variable?
Sentence: “The finding extends the notion that situation-related affective empathy is a central motivator of altruism directed at helping others in need or distress.” It seems over conclusion, because Authors did not other possible motivators for altruism.
Please remove „The limitation of this study…..“ in Discusion part of manuscript.
It is suggested that the authors further emphasize the importance of this research as well as the contribution to the area.
